# Correlation Between SCORE2-Diabetes and Coronary Artery Calcium Score in Patients with Type 2 Diabetes Mellitus: A Cross-Sectional Study in Vietnam

**DOI:** 10.3390/jimaging11050130

**Published:** 2025-04-23

**Authors:** Hung Phi Truong, Hoang Minh Tran, Thuan Huynh, Dung N. Q. Nguyen, Dung Thuong Ho, Cuong Cao Tran, Sang Van Nguyen, Tuan Minh Vo

**Affiliations:** 1Faculty of Medicine, University of Medicine and Pharmacy at Ho Chi Minh City, Ho Chi Minh City 7000, Vietnam; truongphihung2007@ump.edu.vn (H.P.T.); huynh@thuan.org (T.H.); dr.trancaocuong@ump.edu.vn (C.C.T.); vominhtuan@ump.edu.vn (T.M.V.); 2Thong Nhat Hospital, Ho Chi Minh City 7000, Vietnam; dunghothuong@gmail.com; 3Faculty of Medicine, University of Health Sciences, Vietnam National University, Ho Chi Minh City 7000, Vietnam; nnqdung@medvnu.edu.vn; 4Faculty of Medicine, University of Medicine and Pharmacy at Thai Nguyen City, Thai Nguyen City 2500, Vietnam; dr.nguyensang@gmail.com

**Keywords:** SCORE2-Diabetes, Coronary Artery Calcium Score (CACS), Type 2 Diabetes Mellitus (T2DM), cardiovascular risk assessment

## Abstract

(1) Background: The SCORE2-Diabetes model has been developed as an effective tool to estimate the 10-year cardiovascular risk in patients with diabetes. Coronary computed tomography angiography (CCTA) and its derived Coronary Artery Calcium Score (CACS) are widely used non-invasive imaging tools for assessing coronary artery disease (CAD). This study aimed to evaluate the correlation between CACS and SCORE2-Diabetes in patients with T2DM. (2) Methods: A cross-sectional study was conducted from October 2023 to May 2024. We included patients aged 40 to 69 years with T2DM who underwent a coronary multislice CT scan due to atypical angina. The correlation between CACS and SCORE2-Diabetes was analyzed using Spearman’s rank correlation coefficient. (3) Results: A total of 100 patients met the inclusion criteria, including 71 males and 29 females, with a mean age of 61.9 ± 5.4 years. The differences in CACS and SCORE2-Diabetes among different degrees of coronary artery stenosis were statistically significant (*p* < 0.05). A statistically significant but weak positive correlation was observed between CACS and SCORE2-Diabetes across all risk categories, with Spearman’s rank correlation coefficients ranging from 0.27 to 0.28 (*p* < 0.01). (4) Conclusions: Despite the weak correlation between CACS and SCORE2-Diabetes, understanding their relationship and independent associations with disease severity is valuable. The combination of these two tools may warrant investigation in future studies to potentially enhance cardiovascular risk assessment in T2DM patients.

## 1. Introduction

Cardiovascular diseases (CVDs) and diabetes mellitus pose significant global health burdens. CVDs are the leading cause of death worldwide, accounting for 20.5 million deaths in 2021, comprising approximately one-third of all global deaths [1]. Similarly, the Global Burden of Disease Collaborative Network estimated that, in 2021, there were approximately 529 million people living with diabetes, with a global age-standardized prevalence of 6.1% [2]. Diabetes significantly increases the risk of CVDs, and individuals with both conditions experience a life expectancy reduction of up to 15 years compared to those with CVDs alone [3].

Risk stratification is a critical component of CVD prevention in diabetic patients. The Systematic Coronary Risk Evaluation (SCORE) model, developed by the European Society of Cardiology (ESC), has been widely used for estimating 10-year cardiovascular risk [4]. In 2021, the ESC introduced SCORE2 and SCORE2-OP, which provide improved risk estimation for adults aged 40–89 years [5].

Most recently, in May 2023, the ESC introduced SCORE2-Diabetes, incorporating hemoglobin A1c (HbA1c) and estimated glomerular filtration rate (eGFR) to enhance risk estimation for diabetic patients. In addition to SCORE2 and SCORE2-OP, SCORE2-Diabetes serves as a crucial tool for physicians in estimating the 10-year risk of fatal CVDs.

Alongside 10-year cardiovascular risk estimation tools, Coronary Computed Tomography Angiography (CCTA) serves as a crucial non-invasive imaging modality for evaluating coronary artery disease (CAD). CCTA enables a detailed assessment of coronary artery anatomy, the extent of calcification, and the characteristics and severity of atherosclerotic plaque stenosis. Notably, the Coronary Artery Calcium Score (CACS), derived from CT imaging, has been shown to be a valuable predictor of cardiovascular risk and a guide for preventive treatment in patients with type 2 diabetes mellitus (T2DM). Despite the availability of various cardiovascular risk assessment tools, studies examining the correlation between SCORE2-Diabetes and CACS remain limited, particularly in Asian populations, including Vietnam. Therefore, this study aimed to investigate the correlation between CACS obtained through CCTA and SCORE2-Diabetes in patients with T2DM.

## 2. Materials and Methods

This study was approved by the Institutional Review Board of Thong Nhat Hospital, Ho Chi Minh City, from October 2023 to May 2024. All patients enrolled in the study were informed about its purpose, potential risks, and benefits. Written informed consent was obtained before data collection.

Patients aged 40–69 years with T2DM who were referred for a coronary multislice CT scan due to atypical angina were included. Conversely, patients with ECG findings suggestive of ischemic heart disease, a history of percutaneous coronary intervention (PCI), myocardial infarction (MI), or unstable angina, as well as those with arrhythmias, atrial fibrillation (AF), or premature ventricular contractions, were excluded.

### 2.1. Preparation for and Implementation of CT Scan

Before undergoing coronary angiography, patients were required to sign a consent form after receiving a thorough explanation of the procedure. All metal objects, including jewelry and hairpins, had to be removed. Patients were instructed to fast for at least four hours before the scan, with only 50 mL of water permitted.

To ensure optimal imaging quality, the target heart rate for CCTA was maintained below 60 beats per minute, with an upper limit of 65 beats per minute. If the heart rate exceeded this threshold, a beta-blocker, preferably metoprolol (50–100 mg), was administered 1–2 h before the scan. An 18-gauge intravenous catheter was inserted into the right elbow and remained in place for at least 15 min after contrast injection.

The multislice CCTA was performed using the Philips Ingenuity 128 Circular Edition with Omnipaque 350 mg I/mL contrast agent.

### 2.2. SCORE2- DiabetesCalculation

The SCORE2-Diabetes calculation is based on a complex statistical model that incorporates various factors, including age, sex, smoking status, blood pressure, total cholesterol, HDL cholesterol, HbA1c level, and eGFR [6]. In this study, we used the MDCalc mobile application to calculate the SCORE2-Diabetes risk score (https://www.mdcalc.com/calc/10510/score2-diabetes, accessed on 30 January 2025). As per the SCORE2-Diabetes model, cardiovascular risk categories were defined as follows: low risk (<5%), moderate risk (5–10%), high risk (10–20%), and very high risk (≥20%) based on the estimated 10-year cardiovascular risk.

### 2.3. Statistical Analysis

Statistical analysis was conducted using SPSS Statistics version 19. The normality of continuous variables was assessed using the Kolmogorov–Smirnov test. Key continuous variables, including CACS and SCORE2-Diabetes scores, were found to be non-normally distributed (*p* < 0.05); therefore, non-parametric tests were predominantly used. Continuous variables were presented as either mean ± standard deviation (SD) or median, while categorical variables were reported as frequencies and percentages.

For group comparisons, *t*-test and ANOVA were applied to normally distributed continuous variables. If the continuous variables did not follow a normal distribution, the Mann–Whitney U test and the Kruskal–Wallis test were used instead. Categorical variables were analyzed using the Chi-squared test.

To examine the correlation between CACS and SCORE2-Diabetes, Spearman’s rank correlation coefficient was applied. The predictive value of CACS and SCORE2-Diabetes for coronary artery stenosis was assessed using receiver operating characteristic (ROC) curves, and the area under the curve (AUC) was determined. Additionally, the optimal cutoff points for each score in the study population were identified using the Youden index.

## 3. Results

### 3.1. Baseline Characteristics

A total of 100 patients met the inclusion criteria. The mean age was 61.9 years, with females accounting for 29% of the study population. A significant proportion of patients exhibited multiple cardiovascular risk factors, including hypertension (88%) and dyslipidemia (82%). Additionally, 57% were active smokers. The average duration of T2DM among participants was 4.0 years. The demographic and clinical characteristics of the study population are summarized in Table 1.

### 3.2. Coronary Artery Characteristics on Computed Tomographic Angiography

The coronary artery characteristics on CCTA findings demonstrated a significant burden of coronary artery stenosis. Notably, 53% of patients had ≥50% stenosis, indicative of obstructive coronary artery disease (OCAD), while 16% exhibited significant stenosis in the left main coronary artery. The distribution of stenosis among the major coronary arteries was as follows: left anterior descending artery (LAD) in 69% of patients, left circumflex artery (LCx) in 34%, and right coronary artery (RCA) in 47% (Table 2).

Patients with stenosis in the left main (LM), left anterior descending (LAD), left circumflex (LCx), and right coronary artery (RCA) consistently exhibited significantly higher mean calcium scores compared to those with patent arteries (Table 3).

Furthermore, Table 4 highlights a clear relationship between clinical and laboratory data, the SCORE2-Diabetes score, and the severity of coronary artery stenosis. As stenosis severity progresses from patent arteries to non-obstructive coronary arteries (NCAD) and obstructive coronary artery disease (OCAD), the mean SCORE2-Diabetes score also increases significantly. This trend was statistically significant across all comparisons, with *p*-values less than 0.05. Individuals with higher SCORE2-Diabetes scores are more likely to develop severe coronary artery stenosis and, consequently, face a higher risk of adverse cardiovascular events.

### 3.3. Correlation Between Coronary Artery Calcium Score and SCORE2-Diabetes

Across all four risk categories (low, medium, high, and very high), the Spearman correlation coefficient between Coronary Artery Calcium Score (CACS) and SCORE2-Diabetes was positive, ranging from 0.27 to 0.28, indicating a weak positive correlation between the two indices. This correlation was statistically significant in all four risk categories, with *p*-values < 0.05. Specifically, in the low, medium, and high-risk categories, the *p*-value was 0.005, while in the very-high-risk category, it was 0.006. The correlation strength was consistent across risk groups, with a coefficient of approximately 0.28 in the low, medium, and high-risk categories, and 0.27 in the very-high-risk category (Table 5).

CACS values differed significantly among patients with patent vessels, single-vessel disease, and multi-vessel disease (*p* = 0.003). Patients with patent vessels had the lowest mean calcium score (2.1 ± 4.9), indicating minimal calcification. In contrast, those with single-vessel disease exhibited a significantly higher mean score (67.6 ± 182.7), and the score increased substantially in patients with multi-vessel disease (220.0 ± 337.1). This finding suggests a correlation between higher CACS and increased disease severity.

Similarly, CACS classification demonstrated statistically significant trends (*p* < 0.001). Patients with no calcification (score = 0) were more prevalent in the patent vessel group (43.8%) compared to only 28.1% in both the single- and multi-vessel disease groups. Moreover, calcium scores ranging from 101 to 400 and >400 were strongly associated with multi-vessel disease. Notably, 94.1% of patients with calcium scores between 101 and 400 and 80% of those with scores >400 had multi-vessel disease (Table 6).

Furthermore, SCORE2-Diabetes classifications showed significant associations with coronary artery disease severity. The mean SCORE2-Diabetes score for the low-risk category differed significantly among the three groups (*p* = 0.006), with the highest scores observed in patients with multi-vessel disease (13.0 ± 4.3), followed by those with single-vessel disease (10.9 ± 2.8) and patent vessels (10.3 ± 2.9). Similar trends were noted for the moderate-risk (*p* = 0.006), high-risk (*p* = 0.012), and very-high-risk (*p* = 0.025) categories.

For instance, patients classified as very-high-risk exhibited a progressive increase in SCORE2-Diabetes scores, from 27.2 ± 5.7 in the patent vessel group to 30.6 ± 6.4 in the single-vessel disease group and 33.0 ± 9.5 in the multi-vessel disease group. These findings highlight a strong association between higher SCORE2-Diabetes values and the extent of coronary artery disease (Table 6).

CACS demonstrated a sensitivity of 70% and a specificity of 96% for diagnosing coronary artery stenosis, with an optimal cutoff value of 38.4. The difference compared to the random diagnostic threshold was statistically significant (*p* < 0.001).

Meanwhile, the SCORE2-Diabetes score in the low- and intermediate-risk categories shared the same optimal cutoff values of 13.7 and 18.3, respectively, with a sensitivity of 52% and a specificity of 82%. This difference from the random diagnostic threshold was statistically significant (*p* = 0.01). In the high-risk category, the optimal cutoff value for SCORE2-Diabetes was 18.8, yielding a higher sensitivity of 74% but a lower specificity of 60% (*p* = 0.01). For the very high-risk category, the optimal cutoff was 32.5, with a sensitivity of 60% and a specificity of 74%, and this difference was also statistically significant (*p* = 0.02).

These findings suggest that CACS is an effective tool for identifying patients with coronary artery stenosis while also reliably excluding those without the disease (Figure 1).

## 4. Discussion

This cross-sectional study was conducted on 100 patients from October 2023 to May 2024. The study aimed to investigate the association between CACS and SCORE2-Diabetes, a risk assessment tool derived from SCORE2 for estimating the likelihood of fatal and non-fatal cardiovascular disease in individuals with pre-existing diabetes aged 40–69 years. The study population comprised 71 male and 29 female patients, with a mean age of 61.9 ± 5.4 years.

### 4.1. Baseline Characteristics

The study results indicated that patients with T2DM exhibited multiple abnormalities in paraclinical indicators. The average systolic blood pressure was 150.6 ± 16.1 mmHg, exceeding the recommended levels outlined in the 2024 ESC Guidelines for the management of elevated blood pressure and hypertension [7]. This elevated average SBP aligns with the high prevalence of diagnosed hypertension (88%) in the cohort, suggesting blood pressure control requires attention in this population.

Dyslipidemia was also prevalent in this patient population. The mean concentrations of total cholesterol, LDL-Cholesterol, and triglycerides were 4.5 ± 1.5 mmol/L, 2.5 ± 1.2 mmol/L, and 2.3 ± 1.4 mmol/L, respectively, all of which were elevated relative to optimal targets. The mean HDL-cholesterol was relatively low (1.2 ± 0.3 mmol/L), which, combined with elevated triglycerides (2.3 ± 1.4 mmol/L), suggests atherogenic dyslipidemia, common in T2DM [8]. Dyslipidemia is an integral component of metabolic syndrome and is commonly associated with T2DM. In the studied patient population, lipid parameters deviated significantly from recommended values: total cholesterol, LDL-cholesterol, and triglycerides were elevated, while HDL-cholesterol was decreased. This lipid disorder pattern—characterized by elevated triglycerides and reduced HDL-C—is identified as atherogenic dyslipidemia, a common clinical manifestation in patients with T2DM.

The pathogenesis of atherogenic dyslipidemia in T2DM is closely related to insulin resistance. Insulin resistance not only impairs glucose utilization in peripheral tissues but also disrupts lipid metabolism in the liver. Increased hepatic production of triglyceride-rich lipoproteins (VLDL) contributes to elevated blood triglyceride levels. At the same time, decreased efficiency of lipoprotein lipase reduces triglyceride breakdown and impairs HDL turnover, resulting in decreased HDL-C levels. This combination promotes the development of atherosclerosis, a major risk factor for cardiovascular events such as myocardial infarction and stroke. Epidemiological studies have shown that in diabetic patients, effective control of lipid parameters—particularly LDL-C and non-HDL-C—can significantly reduce the incidence of cardiovascular events. Guidelines from cardiovascular associations currently recommend aggressive lipid management in this patient group, including lifestyle modifications and the use of moderate- to high-intensity statins depending on the level of risk. Therefore, the identified lipid disorder pattern in the study population not only reflects the high cardiovascular risk among T2DM patients but also underscores the importance of early screening and timely intervention for dyslipidemia as essential components of comprehensive treatment strategies.

The renal function in the study group showed signs of decline, with an average estimated eGFR of 77.0 ± 13.5 mL/min. While this average is above the threshold for severe CKD stage G3a (<60), it indicates that many patients are below optimal levels (>90) and suggests early renal function decline is prevalent, consistent with diabetes being a leading cause of CKD. Renal impairment is a common complication of diabetes and an independent risk factor for cardiovascular disease. A study by Ene-Iordache et al. demonstrated that the risk of cardiovascular diseases rises as eGFR declines [9].

Furthermore, the high prevalence of active smokers (57%) represents a critical modifiable risk factor likely contributing to the CAD burden observed. Average glycemic control (HbA1c 7.0 ± 1.1%) was near typical targets, but hyperglycemia remains a core issue in T2DM pathogenesis. Urea and creatinine levels were generally within normal ranges, reflected in the mean eGFR.

Overall, the laboratory findings revealed that the study population had multiple cardiovascular risk factors. The coexistence of these factors exacerbates disease burden and may explain the high prevalence of coronary artery lesions observed on coronary computed tomography [10,11].

### 4.2. Coronary Artery Characteristics on Computed Tomographic Angiography

Regarding the severity of coronary artery stenosis, 53% of patients had coronary artery stenosis ≥ 50%. This is higher than the 36% reported in the CONFIRM registry for T2DM patients [12] but lower than the 90.9% in a specific 2021 Chinese subgroup analysis [13], suggesting population variability. This high prevalence underscores the accelerated atherosclerosis in T2DM driven by hyperglycemia, insulin resistance, dyslipidemia, hypertension, and inflammation, leading to endothelial dysfunction and plaque progression [8].

Additionally, 69% of patients had left anterior descending (LAD) artery stenosis, followed by right coronary artery (RCA) stenosis in 47%, left circumflex (LCx) artery stenosis in 34%, and left main (LM) artery stenosis in 16%. Moreover, 53% of patients had stenosis in two or more coronary arteries, including 19% with stenosis in three branches and 7% with stenosis in all four major branches. In contrast, 20% of patients had no stenosis in any coronary artery. The distribution of coronary calcification was as follows: LAD (69%), LCx (34%), RCA (47%), and LM (16%). These findings are consistent with the study by Alluri et al. [14]. The predominance of LAD lesions in patients with type 2 diabetes mellitus (T2DM) can be explained by the pathophysiology of insulin resistance, dyslipidemia, and chronic inflammation—all of which promote the development of atherosclerotic plaques in major coronary arteries with high flow, such as the LAD. Additionally, the LAD is the primary vessel supplying the anterior myocardial wall—a region particularly sensitive to ischemia—so lesions in this artery are associated with worse prognoses and are often a primary target for revascularization. The high rate of multivessel disease (53%), including 7% with stenosis in all four main coronary branches, reflects the diffuse nature of coronary lesions—a hallmark of T2DM. This pattern mirrors the systemic impact of both microvascular and macrovascular complications driven by chronic hyperglycemia, leading to progressive atherosclerosis in multiple arterial territories. Moreover, coronary lesions in T2DM patients tend to be more challenging to treat due to smaller vessel size, extensive calcification, and concomitant microvascular disease, all of which reduce the efficacy of revascularization strategies.

The observation that calcification rates in the LAD, RCA, and LCx mirror stenosis rates suggests that calcification is an imaging marker that correlates well with the severity of coronary lesions in T2DM. However, it is important to note that not all calcified lesions cause hemodynamically significant stenosis. Therefore, in T2DM patients—who are at high risk for cardiovascular events—integrating calcification assessment (via CCTA) with functional indices (such as FFR-CT or iFR) is essential to optimize treatment strategies. Overall, the findings of this study align with the known pattern of coronary artery disease in T2DM: diffuse, multivessel involvement, heavy calcification, and predominance in the LAD. Recognizing the morphological characteristics of coronary lesions in this population is crucial for personalizing treatment, selecting the appropriate intervention strategy, and improving long-term outcomes.In terms of SCORE2-Diabetes indicators (gender, age, smoking status, systolic blood pressure, total cholesterol, HDL-cholesterol, HbA1c, and eGFR) [6], there was a significantly higher smoking rate in patients with NCAD and OCAD. Peak systolic blood pressure also increased progressively with the severity of coronary artery stenosis, while HDL cholesterol levels were significantly lower in patients with NCAD and OCAD [15]. These abnormalities, which have been reported in multiple previous studies, are incorporated into the ESC guidelines for estimating cardiovascular risk in diabetic patients [6,15]

### 4.3. Correlation Between CACS and SCORE2-Diabetes Score

Our study demonstrated a positive correlation between CACS and SCORE2-Diabetes across all risk categories, with Spearman correlation coefficients ranging from 0.27 to 0.28 (*p* < 0.01). This indicates a statistically significant but weak positive association. This weak correlation may be attributed to the characteristics of the study population or the cardiovascular risk scoring tool itself. Since SCORE2-Diabetes was developed based on European populations, differences in physical characteristics, culture, and lifestyle may have influenced its applicability to other populations [6]. Further validation and potential recalibration for Vietnamese populations may be needed.

The study by Raggi et al. demonstrated that the correlation between CACS and cardiovascular risk may be influenced by factors such as age, sex, and diabetes status [15]. Their findings also indicated that CACS had a higher predictive value for cardiovascular risk in patients with diabetes than in those without diabetes, and its effectiveness varied depending on age group and gender [15].

Both the European Society of Cardiology (ESC) and the American Heart Association (AHA) recommend the use of CACS for cardiovascular risk screening and stratification [16]. Additionally, CACS has been identified as an independent predictor of mortality [17,18]. Consistent with previous studies, our findings showed that CACS exhibited high diagnostic performance in detecting coronary artery stenosis ≥ 50% (OCAD), with an area under the curve (AUC) of 0.87 (95% CI: 0.8–0.94, *p* < 0.001) [18,19].

On the other hand, the study by Kasim et al. demonstrated that the SCORE2 score is effective in predicting cardiovascular disease risk in Malaysians. Given that Malaysia and Vietnam both belong to the ASEAN region and share cultural similarities, lifestyles, and comparable levels of economic development, these findings suggest that SCORE2 may also be applicable to the Vietnamese population [20].

Additionally, Paiva et al. proposed incorporating CACS as a risk modifier in the SCORE2 model [21]. The combination of these two tools could enhance the identification of high-risk patients, thereby facilitating more accurate risk stratification and enabling the implementation of appropriate preventive and treatment strategies.

Our study demonstrated a correlation between the SCORE2-Diabetes scale and the severity of coronary artery stenosis in patients with T2DM. The SCORE2-Diabetes score increased progressively with worsening stenosis, from the normal group to <50% stenosis and subsequently to ≥50% stenosis, across all risk categories from low to very high.

Additionally, CACS increased in parallel with the number of stenotic coronary branches, with a statistically significant difference (*p* = 0.003). This finding aligns with the study by Elkhoraiby et al., which examined the SCORE2 scale in non-diabetic patients [22,23]. This relationship can be explained by the fact that patients with T2DM often present with multiple cardiovascular risk factors, including hypertension, dyslipidemia, and obesity. These factors, in combination with chronic hyperglycemia, contribute to endothelial damage and atherosclerosis, ultimately leading to coronary artery stenosis [24].

## 5. Limitations

First, this study was limited by its small sample size (100 patients) and was conducted at a single center, making it less representative of the entire population of T2DM patients in Vietnam.

Second, the SCORE2-Diabetes model was developed and calibrated based on European populations, which may limit its applicability to the Vietnamese population due to differences in ethnic characteristics and cardiovascular risk factors. This may contribute to the observed weak correlation and underscores the need for local validation or recalibration.

Third, this study focused solely on the relationship between SCORE2-Diabetes and CACs. Further research incorporating other imaging modalities, such as percutaneous coronary angiography, cardiac magnetic resonance imaging, and intravascular ultrasound, is necessary to provide a more comprehensive assessment.

Despite these limitations, our study provides valuable initial data on the relationship between SCORE2-Diabetes and CACS in Vietnam, a region where research on this topic may be limited. It serves as a baseline for larger, longitudinal, and multi-center studies needed to refine cardiovascular risk assessment in this population.

## 6. Conclusions

This study demonstrated a statistically significant but weak correlation between CACS and SCORE2-Diabetes. Despite this, the combined use of both tools may enhance cardiovascular risk stratification in patients with T2DM. Future studies should focus on the following: (1) validation in larger, multi-center Vietnamese cohorts; (2) longitudinal follow-up to assess the prognostic accuracy of both tools for cardiovascular events; (3) investigating the incremental predictive value of combining CACS with SCORE2-Diabetes; and (4) exploring potential recalibration of SCORE2-Diabetes for optimal performance in this population.

## Figures and Tables

**Figure 1 jimaging-11-00130-f001:**
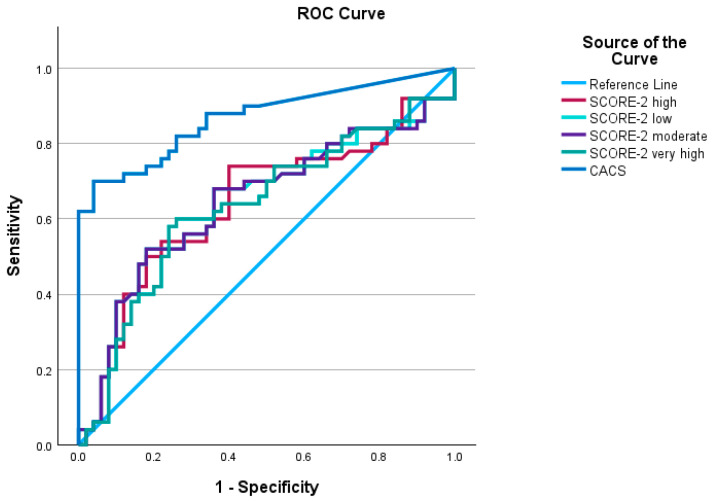
Roc curve of the diagnostic performance of coronary artery stenosis ≥ 50% of Coronary Artery Calcium Score and SCORE2-Diabetes by risk class.

**Table 1 jimaging-11-00130-t001:** The baseline characteristics of all patients.

	Patients (n = 100)
Mean Age (years)	61.9 ± 5.4
Male (n, %)	71 (71.0)
Female (n, %)	29 (29.0)
Hypertension (n, %)	88 (88.0)
Blood pressure (mmHg)	150.6 ± 16.1
Dyslipidemia (n, %)	82 (82.0%)
Active smoking (n, %)	57 (57.0%)
Mean Age of T2DM Diagnosis (years)	58.0 ± 5.6
Duration of T2DM (years)	4.0 ± 3.3
Glucose (mmol/L)	7.7 ± 2.6
HbA1c (%Hb)	7.0 ± 1.1
Ure (mmol/L)	5.4 ± 1.5
Creatinin (µmol/L)	88.2 ± 16.4
eGFR (mL/min)	77.0 ± 13.5
Total Cholesterol (mmol/L)	4.5 ± 1.5
HDL-Cholesterol (mmol/L)	1.2 ± 0.3
LDL-Cholesterol (mmol/L)	2.5 ± 1.2
Triglycerid (mmol/L)	2.3 ± 1.4

HbA1c: Hemoglobin A1c; eGFR: estimated Glomerular Filtration Rate; T2DM: type 2 diabetes mellitus.

**Table 2 jimaging-11-00130-t002:** Distribution of coronary artery stenosis on CCTA.

CCTA Feature		Patients (n = 100)
CCTA	Patent	20 (20.0%)
	<50% (NCAD) (n, %)	27 (27.0%)
	≥50% (OCAD) (n, %)	53 (53.0%)
Coronary artery stenosis	LM (n, %)	16 (16.0%)
	LAD (n, %)	69 (69.0%)
	LCx (n, %)	34 (34.0%)
	RCA (n, %)	47 (47.0%)
Number of stenotic coronary artery branches	1 (n, %)	27 (27.0%)
	2 (n, %)	27 (27.0%)
	3 (n, %)	19 (19.0%)
	4 (n, %)	7 (7.0%)

CCTA: Coronary Computed Tomography Angiogram; Patent: No coronary artery stenosis found. NCAD: Non-Obstructive Coronary Artery Disease; OCAD: Obstructive Coronary Artery Disease; LM: Left Main; LAD: Left Anterior Descending; LCx: Left Circumflex Coronary Artery; RCA: Right Coronary Artery.

**Table 3 jimaging-11-00130-t003:** Coronary Artery Calcium Score in coronary branches.

		Patients (n = 100)	Coronary Artery Calcium Score
LM	Stenosis	16	251.6 ± 210.5
	Patent	84	113.1 ± 284.4
LAD	Stenosis	69	181.6 ± 307.8
	Patent	31	32.1 ± 154.2
LCX	Stenosis	34	257.0 ± 393.1
	Patent	66	72.6 ± 165.1
RCA	Stenosis	47	232.3 ± 359.4
	Patent	53	49.2 ± 128.5

LM: Left Main; LAD: Left Anterior Descending; LCx: Left Circumflex Coronary Artery; RCA: Right Coronary Artery.

**Table 4 jimaging-11-00130-t004:** Relationship between clinical and laboratory data, the SCORE2-Diabetes score, and the severity of coronary artery stenosis.

Feature			Patent (n = 19)	NCAD (n = 31)	OCAD (n = 50)	*p*-Value
Age (years)			61.2	62.6	61.7	0.633
Gender	Male (n, %)		13 (18.3%)	19 (26.8%)	39 (54.9%)	0.263
	Female (n, %)		6 (20.7%)	12 (41.4%)	11 (37.9%)	
Hypertension (n, %)			15 (17.0%)	28 (31.8%)	45 (51.1%)	0.402
Blood pressure (mmHg)			143.7 ± 12.6	148.1 ± 12.8	154.8 ± 18.0	0.019
Smoking (n, %)			6 (11.3%)	13 (24.5%)	34 (64.2%)	0.008
Dyslipidemia (n, %)			17 (20.7%)	25 (30.5%)	40 (48.8%)	0.640
Age of T2DM diagnosis (years)			57.1 ± 4.3	58.7 ± 5.5	57.8 ± 6.2	0.585
Glucose (mmol/L)			7.8 ± 2.5	7.7 ± 2.8	7.6 ± 2.5	0.95
HbA1c (%Hb)			6.8 ± 0.9	7.1 ± 1.1	7.2 ± 1.1	0.38
Ure (mmol/L)			5 ± 0.9	5.6 ± 1.8	5.5 ± 1.5	0.312
Creatinin (µmol/L)			91.5 ± 14.5	89.2 ± 20.3	86.3 ± 14.4	0.46
eGFR (mL/min)			71.9 ± 14.4	74.9 ± 12.4	80.3 ± 13.1	0.036
Total Cholesterol (mmol/L)			4 ± 0.6	4.8 ± 1.3	4.5 ± 1.7	0.240
HDL-Cholesterol (mmol/L)			1.1 ± 0.3	1.3 ± 0.3	1.1 ± 0.2	0.027
LDL-Cholesterol (mmol/L)			1.9 ± 0.6	2.8 ± 1.1	2.5 ± 1.4	0.048
SCORE2-Diabetes Low risk			10.1 ± 2.9	11.6 ± 3.5	12.8 ± 4.1	0.027
SCORE2-Diabetes Moderate risk			13.3 ± 4	15.3 ± 4.9	17.1 ± 5.8	0.027
SCORE2-Diabetes High risk			17 ± 4.7	20.6 ± 7.2	21.9 ± 7.1	0.032
SCORE2-Diabetes Very high risk			26.9 ± 5.7	31.4 ± 8.9	32.6 ± 8.5	0.038
CACS			2.2 ± 5.1	14.6 ± 18.3	260.6 ± 351.4	<0.001
CAC classification		0	13 (40.6%)	14 (43.8%)	5 (15.6%)	<0.001
		1–100	6 (14.6%)	17 (41.5%)	18 (43.9%)	
		101–400	0 (0.0%)	0 (0.0%)	17 (100.0%)	
		>400	0 (0.0%)	0 (0.0%)	10 (100.0%)	

HbA1c: Hemoglobin A1c: eGFR: estimated Glomerular Filtration Rate; CACS: Coronary Artery Calcium Score; CAC: Coronary Artery Calcium.

**Table 5 jimaging-11-00130-t005:** Correlation between Coronary Artery Calcium Score and SCORE2-Diabetes.

Spearman’s rho	Spearman’s Rank Correlation Coefficient	*p*-Value
SCORE2-Diabetes Low risk	0.28	0.005
SCORE2-Diabetes Moderate risk	0.28	0.005
SCORE2-Diabetes High risk	0.28	0.005
SCORE2-Diabetes Very high risk	0.27	0.005

**Table 6 jimaging-11-00130-t006:** Correlation between higher SCORE2-Diabetes and the extent of coronary artery disease.

Feature		Patent (n = 20)	Single-Vessel Disease (n = 27)	>1 Vessel Disease (n = 53)	*p* Value
CACS		2.1 ± 4.9	67.6 ± 182.7	220.0 ± 337.1	0.003
CACS classification	0	14 (43.8%)	9 (28.1%)	9 (28.1%)	<0.001
	1–100	6 (14.6%)	15 (36.5%)	20 (58.8%)	
	101–400	0 (0.0%)	1 (5.9%)	16 (94.1%)	
	>400	0 (0.0%)	2 (20.0%)	8 (80.0%)	
Coronary artery stenosis	Patent	19 (100.0%)	0 (0.0%)	0 (0.0%)	<0.001
	<50%	1 (3.2%)	17 (54.8%)	13 (41.9%)	
	≥50%	0 (0.0%)	10 (20.0%)	40 (80.0%)	
** *SCORE2 Diabetes* **					
SCORE2-Diabetes Low risk		10.3 ± 2.9	10.9 ± 2.8	13.0 ± 4.3	0.006
SCORE2-Diabetes Moderate risk		13.5 ± 4.1	14.4 ± 4.0	17.4 ± 6.0	0.006
SCORE2-Diabetes High risk		17.3 ± 4.7	19.5 ± 5.0	22.3 ± 8.0	0.012
SCORE2-Diabetes Very high risk		27.2 ± 6.0	30.6 ± 6.4	33.0 ± 9.5	0.025

CACs: Coronary Artery Calcium Score.

## Data Availability

The original data of this study will be made available upon reasonable request from relevant researchers (link: https://docs.google.com/spreadsheets/d/1MrzxKCDZxUnknXerNY7NCFxYSZINpvpi/edit?gid=1625321709#gid=1625321709, accessed on 15 March 2025).

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
