# Peer review of "Correlation Between SCORE2-Diabetes and Coronary Artery Calcium Score in Patients with Type 2 Diabetes Mellitus: A Cross-Sectional Study in Vietnam"

_2313-433X, 2025, doi:10.3390/jimaging11050130_

Round 1
Reviewer 1 Report
Comments and Suggestions for Authors
This paper attempts to address the limitations of existing studies on the correlations between type 2 diabetes mellitus (T2DM) and cardiovascular diseases (CVDs), particularly in the context of Vietnam. I appreciate the authors' efforts to fill this gap by involving 100 T2DM patients and examining their clinical characteristics in relation to CVDs. However, the authors did not provide a detailed discussion on the reasons behind their findings. Additionally, the introduction section lacks sufficient clarity and narrative structure. Overall, the manuscript requires significant revisions before it can be considered for potential publication.
- Abstract: The abstract does not follow the normal format of an abstract. Please ensure that the formatting complies with the journal’s guidelines.
- Introduction, Line 52: Given that 2019 data is now six years old, the authors should consider citing more recent statistics to ensure the study reflects current trends.
- Introduction: This section appears overly brief and would benefit from a more comprehensive narrative structure. The authors should begin by introducing cardiovascular diseases (CVDs) and type 2 diabetes mellitus (T2DM), highlighting how both conditions individually and jointly pose significant health risks. This would establish a clear rationale for the need for preventive measures. Following this, the authors can transition into emphasizing the specific focus of their research—investigating the correlation between CVD-related conditions and diabetes. In Line 75, the authors mention that correlation studies remain challenging due to limited existing research. To strengthen this point, they should cite relevant existing studies and discuss the limitations of these studies in greater detail. This would provide a stronger justification for their research and support their claim that more studies are needed in the Asian context, particularly in Vietnam. The authors should also explicitly explain why findings from existing studies cannot be directly applied to the Asian population. These reasons—such as genetic, environmental, lifestyle, or healthcare system differences—should be clearly stated in the Introduction to establish the research gap more convincingly.
- Materials and Methods: I suggest that the authors present their methodological framework in the form of a flowchart for better clarity.
- Table 1: Does the "Age of diagnosis T2DM" in row 8 refer to the mean age? Additionally, the table lacks clarity. There is no explanation of the source of the clinical values — are they representative of all patients? Since the study includes a total of 100 patients, it is unclear why male participants are not included in the table. Furthermore, the "Age" in row 2 should be specified as "Mean Age" for better clarity.
- Line 135: There is a spelling mistake: "characteristics" is misspelled.
- Table 2: Is “Patent” correct?
- Discussion, Line 262: Please verify the duration of the study. It was previously stated that the study was conducted from October 2023 to May 2024 (Line 80).
- Line 265: Again, please verify the age group. It was previously stated that the participants were between 40 and 70 years old (Line 83).
- Line 269-272: Given that hypertension is defined by elevated blood pressure levels, the authors should explore the direct relationship between the recorded blood pressure measurements and the diagnosis of hypertension in the study population. A statistical analysis or correlation between blood pressure levels and hypertension status would strengthen this discussion and provide clearer insights into the findings.
- Line 273-277: The authors should address the findings related to HDL-cholesterol, as no commentary is provided on this aspect. Furthermore, a more detailed discussion of dyslipidemia in relation to total cholesterol, LDL-cholesterol, and triglycerides is needed to provide a more comprehensive understanding of these lipid abnormalities.
- Line 278: What is the normal reference range for eGFR? The authors state that the renal function in this age group showed signs of decline — however, it is unclear from which value to which value this decline occurred. The statement lacks clarity and requires substantial clinical references to support the claim.
- Section 4.1.: The clinical features such as smoking status, glucose levels, HbA1c, urea, and creatinine are mentioned in the table but are not addressed in the discussion. The authors should provide commentary on these variables to offer a more comprehensive analysis of the study findings.
- Line 288-290: A significant portion of the patients exhibited strong coronary artery stenosis. However, the authors failed to discuss the potential underlying causes of this finding in greater detail. Specifically, what is the relationship between diabetic conditions and cardiovascular diseases (CVDs)? The authors should explore this association further, as it is a critical aspect of the study.
- Line 294-296: The study reports that 69% of patients had stenosis in the left anterior descending (LAD) artery, followed by right coronary artery (RCA) stenosis in 47%, left circumflex (LCx) artery stenosis in 34%, and left main (LM) artery stenosis in 16%. The authors should address why this particular distribution of stenosis is observed. Are there underlying factors or clinical characteristics that explain the higher prevalence of LAD artery involvement compared to the other coronary arteries?
- Line 298-301: The authors mention that the findings are consistent with the study by Alluri et al. [14]. However, they should explicitly compare their results with those of Alluri et al. and discuss any similarities or differences in more detail. Additionally, it would strengthen the manuscript if the authors could cite one or two more relevant studies to further support their findings.
- Section 5: I appreciate the authors' transparency regarding the limitations of their study. However, they should further justify how these limitations influenced the design of the study. Specifically, the authors should explain that despite these limitations, the study was conducted to address important gaps in the literature. Additionally, the authors could emphasize that the initial findings from this study are intended to serve as a baseline for future, more extensive studies on similar topics. This is especially beneficial in a country like Vietnam, where research on this topic may be limited, and these findings could help guide future healthcare strategies.
- Conclusions: This section would benefit from a more detailed discussion on future research directions. The authors should outline potential avenues for further investigation, such as the exploration of larger, multi-center studies, or the long-term validation of SCORE2-Diabetes in diverse populations - that future studies should assess how well SCORE2-Diabetes continues to predict cardiovascular risk over time and whether it remains effective across different patient groups. Expanding on these future steps would help strengthen the impact and applicability of the study’s findings.
- Please review the manuscript for grammatical errors and ensure that the formatting complies with the journal's guidelines.
Please check the English throughout the manuscript.
Reviewer 2 Report
Comments and Suggestions for Authors
Reviewer opinion
- Section 2.3 Statistical Analysis: “The normality of continuous variables was assessed using the Kolmogorov-Smirnov’s test.” -> There is no corresponding description in the results.
- Table 5: the description in the middle column should be “ Spearman’s rank correlation coefficient”? not “CACS”?
- Conclusion: “Despite this, the combined use of both tools may enhance cardiovascular risk stratification in patients with type 2 diabetes.” There is no data about the “combined use” in this study. Those scores were only individually tested for AUCROC and showed CACS is better. Or maybe you might try to construct a new score incorporating CACS and SCORE2-Diabetes? Sincere they are weakly associated, its possible to get a higher AUC value of ROC.
- Why only test them in OCAD(>50% stenosis) vs non-OCAD? How about test them on patient vs non-patent too?
- SCORE2-Diabetes are created to assess the risk of CVD in 10 years, and CACS to classify the calcification status. How about use these raw data to design a new score by logistic regression to identify OCAD or other clinical important outcome?
- This study aimed to evaluate the correlation between CACS and SCORE2-32 Diabetes in patients with T2DM. But what’s the clinical application or need to know the correlation? Is it important to know?

Reviewer 3 Report
Comments and Suggestions for Authors
This cross-sectional study investigates the association between SCORE2-Diabetes, a newly developed cardiovascular risk model tailored for diabetic patients, and the Coronary Artery Calcium Score (CACS), an established non-invasive marker of coronary artery disease (CAD). The study enrolled 100 Vietnamese patients with type 2 diabetes mellitus (T2DM) who underwent coronary CT angiography. The authors assessed the correlation between SCORE2-Diabetes and CACS, stratified by CAD severity. A statistically significant but weak positive correlation (Spearman’s rho ≈ 0.27–0.28) was found, and both metrics were positively associated with CAD burden. The findings suggest that combining these tools could enhance cardiovascular risk stratification in diabetic populations.
The manuscript is generally readable and structurally sound, but several grammatical and stylistic issues detract from its clarity and professionalism. As such the presented material qualifies to be published in J. Imaging. However, a few questions remain, and clarifications and corrections should be applied in a revised version of the manuscript prior to publication:
MA:
1) Small, Single-Center Sample Size: The study’s limited sample size (n=100) and single-center design in Vietnam restrict generalizability to broader T2DM populations, especially across different ethnicities or healthcare settings.
2) Limited External Validity of SCORE2-Diabetes: SCORE2-Diabetes was calibrated using European populations. Its application to Asian cohorts—especially in Vietnam—may not be fully valid due to differences in lifestyle, genetics, and healthcare systems.
Minor:
- Line 79: Redundancy: “This cross-sectional study was conducted with the approval…” correct as “This study was approved by the Institutional Review Board…”
- “characrteristics” correct as “characteristics”.
Round 2
Reviewer 1 Report
Comments and Suggestions for Authors
I thank the authors for their response.